# Environmental and nutrition impact of achieving new School Food Plan recommendations in the primary school meals sector in England

Kremlin Wickramasinghe,[1] Mike Rayner,[1] Michael Goldacre,[2] Nick Townsend,[1] Peter Scarborough[1]

[1]Nuffield Department of Population Health, British Heart Foundation Centre on Population Approaches for Non-Communicable Disease Prevention, University of Oxford, Oxford, UK
[2]Unit of Health Care Epidemiology, Nuffield Department of Population Health, University of Oxford, Oxford, UK

**Correspondence to**
Dr Kremlin Wickramasinghe; kremlin.wickramasinghe@dph.ox.ac.uk

## ABSTRACT

**Objectives:** The aim of this modelling study was to estimate the expected changes in the nutritional quality and greenhouse gas emissions (GHGEs) of primary school meals due to the adoption of new mandatory food-based standards for school meals.

**Setting:** Nationally representative random sample of 136 primary schools in England was selected for the Primary School Food Survey (PSFS) with 50% response rate.

**Participants:** A sample of 6690 primary students from PSFS who consumed school meals.

**Outcome measures:** Primary School Food Plan (SFP) nutritional impact was assessed using both macronutrient and micronutrient quality. The environmental impact was measured by GHGEs.

**Methods:** The scenario tested was one in which every meal served in schools met more than half of the food-based standards mentioned in the SFP (SFP scenario). We used findings from a systematic review to assign GHGE values for each food item in the data set. The GHGE value and nutritional quality of SFP scenario meals was compared with the average primary school meal in the total PSFS data set (pre-SFP scenario). Prior to introduction of the SFP (pre-SFP scenario), the primary school meals had mandatory nutrient-based guidelines.

**Results:** The percentage of meals that met the protein standard increased in the SFP scenario and the proportion of meals that met the standards for important micronutrients (eg, iron, calcium, vitamin A and C) also increased. However, the SFP scenario did not improve the salt, saturated fat and free sugar levels. The mean GHGE value of meals which met the SFP standards was 0.79 (95% CI 0.77 to 0.81) kgCO$_2$e compared with a mean value of 0.72 (0.71 to 0.74) kgCO$_2$e for all meals. Adopting the SFP would increase the total emissions associated with primary school meals by 22 000 000 kgCO$_2$e per year.

**Conclusions:** The universal adoption of the new food-based standards, without reformulation would result in an increase in the GHGEs of school meals and improve some aspects of the nutritional quality, but it would not improve the average salt, sugar and saturated fat content levels.

## Strengths and limitations of this study

- This study shows a method to quantify simultaneously the nutritional impact and carbon footprint of policy options which can have an impact on our diets. To the best of the authors' knowledge this is the first study to estimate greenhouse gas emissions (GHGEs) change as a result of a national food policy. Most of the previous studies compare the GHGE of different dietary patterns and their health outcomes.
- Dietary data come from the largest primary school meals survey in England, the Primary School Food Survey (PSFS) conducted in 2009.
- The GHGEs estimates come from a systematic review conducted by authors (for a previous study). Since GHGE per food item in this study comes from multiple sources, the range in GHGE values for food items would be larger than from studies taking them from a single source. These reported GHGE values are more acceptable given that it is very difficult to pick a single study to best represent the GHGE values for a range of different sources of food.
- This study estimated the GHGE of food items served to the plate and does not deal with the GHGE of leftover food and food waste.
- In this nationally representative data set, we have data only for a single meal of the day, per child and it is difficult to draw conclusions about 'healthiness of the overall diets' using common tools, which require daily or weekly intakes.
- This is a modelling study and does not present results of an evaluation.

## INTRODUCTION

Schools are an important partner in population-level nutrition promotion.[1] Over the past few decades there have been many policies to promote diet and physical activity in schools.[2] Recent policies have addressed the issue of sustainability of school food in addition to their nutritional quality.[3]

School meals make a substantial contribution to greenhouse gas emissions (GHGEs) in the UK.[4] GHGE reduction is a priority concern globally which requires actions from all sectors.[5] Despite various policy actions, globally the total anthropogenic GHGEs have continued to increase from 2000 to 2010.[6] The food sector is one of the main sources of GHGEs and to achieve meaningful reductions in GHGEs, we need to address both production and consumption of food.[7–9] Tackling both climate change impact and health impacts of food is widely discussed as important policy priorities globally.[10–13] There is a growing literature looking at the GHGE of diets and their nutritional quality or health impacts.[11 14–17]

The UK Government introduced a policy on universal free school meals for children in reception, years 1 and 2 (4–6 years old) in state-funded schools in England from September 2014.[18] Official figures show that within 3 months of introducing this policy 1.3 million more children started eating free school meals, resulting in an uptake close to 85% of all infants.[19] Several policies have been implemented to improve the nutritional standards of school meals[20] and some policies such as the Government food procurement policy or Government buying standards have addressed sustainability standards of food.[21] One of the most recent changes to school meals in England was introduced by the School Food Plan (SFP).[22] It provided a series of action points for head teachers to improve the quality of school meals. Based on the suggestions of an expert panel appointed to review school food standards in the UK, the SFP concluded that nutrient-based standards were too difficult to interpret and school meal guidelines should be based only on food-based standards.[23] A report of the Children's Food Trust had found that the introduction of nutrient-based standards had improved school meals in England.[24] There has long been a division of opinion among nutrition experts regarding whether food-based standards are more beneficial than nutrient-based standards.[25] A new set of food-based standards (instead of nutrient-based standards) was published by the Department for Education in June 2014. Since September 2014, schools are now legally required to provide meals that comply with these standards.[18] The impact of these food-based standards is also to improve the nutritional values of school meals.

The SFP mentions the sustainability of food as an important issue to address.[22] It recommends that schools learn from each other how to promote local and sustainable food. These are measures widely promoted to reduce the environmental impact of diets.[26] It is important to quantify the changes in GHGEs that would result from the large changes to school food demanded by the SFP. Previous work has shown that improving the nutritional quality of meals (the primary aim of the SFP) does not guarantee that the GHGEs of the meal will be reduced.[16 27] The nutritional quality or the healthiness of a primary school meal can be measured using several methods. This paper adopts two such methods, explained in the Methods section.

The aim of this modelling study is to estimate the expected changes in the nutritional quality and GHGEs of primary school meals due to the adoption of new mandatory food-based standards for school meals.

## METHODS

This study used the 'Primary School Food Survey 2009 (PSFS)' data set, a nationally representative survey in 139 primary schools in England.[24] The study involved 6690 students who ate school meals and 3488 students who brought packed lunches. This survey was commissioned and funded by the Children's Food Trust (formerly the School Food Trust).

The unit in the PSFS data set is a food item (ie, individual foods that combine to form a meal, eg, 'spaghetti' and 'bolognese sauce'). Nutritional information for all the foods available to the student on that day and food items chosen by a sample of students was recorded in the PSFS data set. Nutritional content of the foods was estimated with reference to the Food Standards Agency Nutrient Databank ( FSA. NDNS Nutrient Databank version 1.32.0 London: Food Standards Agency, 2002.) In the PSFS data set there were 1556 unique food items consumed by students.

We estimated the GHGE values for each food item in the PSFS database. There are two scenarios used for comparison in this paper. They are the SFP scenario and the pre-SFP scenario. Based on the new SFP, we developed criteria to identity school meals in the PSFS data set which met those food-based standards, applicable to a daily primary school meal. These meals were grouped to estimate the average nutritional and GHGE values of the 'SFP scenario'. The average values for all meals in the PSFS data set provide the 'pre-SFP' scenario. Comparison with the average values is more suitable, as we aim to estimate the changes to nutritional quality and GHGEs if all primary school meals adopt the new SFP standards.

## GHGE DATA ON PRIMARY SCHOOL MEALS

We conducted a systematic review to estimate the GHGEs associated with the production of 100 g of different food groups. This systematic review involved searching peer-reviewed and the grey literature from 1995 to 2012 on GHGEs associated with production and consumption of food items (estimated by Life Cycle Analysis). This systematic review provides GHGE values for food items listed in a commonly used Food Frequency Questionnaire (FFQ).[4 28] We allocated GHGE values to food items in the PSFS database using an approach which was adopted from a previous study conducted by Scarborough et al.[29]

More details about the PSFS database and allocation of GHGE values are described in a previous paper.[4]

## SFP MEALS

The SFP provides a list of food-based standards under six main food groups: (1) starchy foods, (2) fruit and vegetables, (3) milk and diary, (4) foods high in fat, sugar and salt, (5) healthier drinks and (6) meat, fish, eggs and beans. Under each category there are standards which apply daily and some standards which apply weekly. For example, in the starchy foods category a daily standard is 'one or more portions from this category daily' and a weekly standard is 'starchy food cooked in oil or fat, not more than 2 days per week'.

In identifying the meals in the PSFS that meet the SFP standards, it was not possible to use every single SFP standard. The PSFS data set is the best available data source for school meals in England, but it surveyed each child for 1 day only, this means that it does not have data on weekly diets. Therefore, we only applied standards which should be adhered by all meals on a daily basis. The survey did collect information on all the food items available in the school for a week. But the aim of this study was to analyse individual meals consumed by children and not the nutritional quality of food available to children.

Table 1 shows the standards which are applicable to each meal. There is at least one standard from each food group. Portion sizes were given in the SFP document for each food group.[23]

The PSFS database provides the weight of each food item. Based on the portion size provided by the SFP, a variable was created to define whether each food item contained the required amounts according to the guidelines. Per each meal, a new set of variables were created to identify whether it met each of the standards listed in table 1.

There are seven daily standards and an equal weighting was given to each of them in order to allocate a score out of 7 for each meal. In this analysis, achieving more than half of the total score (4 or more) was used to classify 'SFP scenario meals' and to compare both the nutritional quality and the GHGEs associated with these meals in comparison with current primary school meals. This bench mark of achieving half of the food-based standards were used because only one meal in the data set achieved all the standards and previous studies which used the PSFS data set achieved half of nutrient-based standards to define healthy meals.[4]

## DEFINING HEALTHY AND UNHEALTHY MEALS

The identification of 'healthy' school meals is not straightforward. Two nutrient-based definitions were used to define a 'healthy school meal' in this paper. The first is based on the three nutrients that are of greatest public health concern: saturated fat, non-milk extrinsic sugars (NMES) and salt.[30] The second is based on a list of 14 nutrients (shown in table 2) used by the PSFS to quantify the nutritional quality of school meals.[30] While recognising that a single meal cannot achieve all 14 nutrient-based standards the second method is based on achieving any 7 or more standards out of 14 nutrient-based standards set for primary school meals before the introduction of new food-based standards.[31] This method considers both healthy and unhealthy nutrients with equal weights.

The data set was created in Microsoft Excel and it was imported in to STATA V.11SE for the analysis.

## RESULTS

In the data set 42% of meals met the standard for having one or more portions of vegetables. One in four achieved the recommendation for fruit (note that fruit was offered in more than two-thirds of the daily menus, but this analysis shows that only 25% of students ate fruit in the portion size specified by the SFP). Drinking water is the recommendation that the highest percentage of meals met (67%) and more than half of the meals met the recommendation for not having any confectionery or chocolate (table 1).

| | Food | Portion size (g) aged 4–10 years | Per cent of current school meals achieved |
|---|---|---|---|
| 1 | One or more portions of vegetables | 40–60 | 42.0 |
| 2 | One or more portions of fruit | 40–60 | 25.5 |
| 3 | A portion of milk and dairy | Milk: 150–200 Cheese: 20–30 Yoghurt: 80–120 | 12.9 |
| 4 | One or more portions of starchy food | Potatoes: 120–170 Bread: 50–70 | 16.7 |
| 5 | A portion of meat, fish, egg or alternatives | Red meat: 50–80 White meat: 60–85 Fish: 60–90 Eggs: 1 egg | 18.2 |
| 6 | Drinking water | Always | 67.0 |
| 7 | No confectionery or chocolate | None | 50.4 |

Table 1   New food-based standards for a primary school meal and the percentage of meals achieving them

**Table 2** Nutrient based standards for a primary school meal; mean, SE and percentage of meals achieved each standards—pre-SFP and SFP scenario

| | Nutrient | Minimum/maximum | Values aged 4–10 years | Pre-SFP meals (n=6691) | | | SFP scenario meals (n=2879) | | |
|---|---|---|---|---|---|---|---|---|---|
| | | | | Mean | SE | Per cent of meals achieved | Mean | SE | Per cent of meals achieved |
| 1 | Energy (kcal) | | 504–557 | 485.95 | 2.19 | 11.1 | 522.9 | 3.46 | 12.33 |
| 2 | Protein (g) | Minimum | 7.5 | 18.47 | 0.09 | 96.0 | 19.56 | 0.52 | 97.36 |
| 3 | Carbohydrate (g) | Minimum | 70.6 | 70.45 | 0.33 | 45.8 | 75.91 | 0.52 | 53.66 |
| 4 | Non-milk extrinsic sugars (g) | Maximum | 15.5 | 14.06 | 0.14 | 60.1 | 15.10 | 0.23 | 55.85 |
| 5 | Fat(g) | Maximum | 20.6 | 16.26 | 0.11 | 72.8 | 17.63 | 0.17 | 67.84 |
| 6 | Saturated fatty acids(g) | Maximum | 6.5 | 6.15 | 0.05 | 60.6 | 6.61 | 0.07 | 55.02 |
| 7 | Fibre(g) | Minimum | 4.2 | 4.76 | 0.03 | 54.7 | 5.18 | 0.04 | 63.67 |
| 8 | Sodium (mg) | Maximum | 499 | 528.71 | 3.74 | 54.0 | 558.29 | 5.85 | 50.43 |
| 9 | Vitamin A (µg) | Minimum | 175 | 332.79 | 4.95 | 50.8 | 394.89 | 8.24 | 58.67 |
| 10 | Vitamin C (mg) | Minimum | 10.5 | 23.05 | 0.24 | 72.1 | 24.67 | 0.34 | 76.3 |
| 11 | Folate (µg) | Minimum | 53 | 63.19 | 0.35 | 59.0 | 67.31 | 0.51 | 66.24 |
| 12 | Calcium (mg) | Minimum | 193 | 202.38 | 1.63 | 44.4 | 210.55 | 2.44 | 48.07 |
| 13 | Iron (mg) | Minimum | 3 | 2.34 | 0.01 | 21.1 | 2.55 | 0.01 | 27.30 |
| 14 | Zinc (mg) | Minimum | 2.5 | 2.14 | 0.01 | 28.9 | 2.31 | 0.02 | 34.53 |

SFP, School Food Plan.

The aim of the new SFP food-based standards is to help schools and students to improve the nutritional quality of school lunches. The percentage of meals that met the protein standard has slightly increased in the SFP scenario, indicating that if the new SFP were adopted by all meals, protein levels in food would increase. Similarly the proportion of meals meeting the standards for important micronutrients (eg, iron, calcium, vitamin A and C) also increased slightly. However, the SFP scenario showed a decline in the proportion of meals achieving fat, saturated fat, sugar and sodium criteria indicating that foods that meet the new SFP guidelines are less likely to meet these nutrient-based criteria (table 2).

There are 2879 meals which met the SFP standards. When salt, saturated fat and NMES were used to define a 'healthy meal' almost 32% of pre-SFP meals were classified as 'healthy'. Out of the meals which met SFP criteria only 27% of meals met these three standards (table 3).

If we apply the method of achieving any seven or more nutrient-based criteria to define 'healthy meals', around 73% of SFP scenario meals could be classified as healthy. In the pre-SFP group around 65% of meals met seven or more criteria. Therefore, if we used this

method to measure the 'healthiness' of school meals it shows that in the SFP scenario more meals are likely to be classified as healthy, compared with pre-SFP meals.

The mean GHGE value of meals which met the SFP criteria is 0.79 (0.77 to 0.81) $kgCO_2e$ compared with 0.72 (0.71 to 0.74) $kgCO_2e$ of a pre-SFP meal. If we assume the total number of primary school children taking the school lunches would remain the same, shifting to a SFP scenario would increase the total emissions by 22 000 000 $kgCO_2e$ per year, which is almost 10% higher than previous primary school meal emissions (table 4).

The difference in the mean GHGEs in the SFP and pre-SFP scenarios could be associated with the amount of food items included in the meal, which makes it more likely to achieve food-based SFP standards and therefore with the total energy of the meal. To test this, we estimated the GHGE per set amount of energy in a primary school meal. The mid-point of the energy reference range for primary school meals is 530 kcal. We estimated the GHGE for a 530 kcal meal in SFP scenario and in a meal in the pre-SFP scenario.

When GHGE per 530 kcal is estimated in SFP and pre-SFP meals, the SFP meals still have slightly more GHGEs 0.84 (0.81 to 0.86) compared with the pre-SFP

**Table 3** Number and percentage of 'healthy' meals, all school meals and SFP scenario

| Definition of 'healthy' meals | Pre-SFP meals (n=6691) | | SFP scenario meals (n=2879) | |
|---|---|---|---|---|
| | Number of meals | Percentage | Number of meals | Percentage |
| Meeting salt, saturated fat and NMES standards | 2139 | 31.97 | 793 | 27.53 |
| Meeting 7 out of all 14 nutrient-based standards | 4312 | 64.45 | 2103 | 73.05 |

NMES, non-milk extrinsic sugars; SFP, School Food Plan.

**Table 4** Mean GHGE of school meals, pre-SFP meals and SFP scenario

| | Mean GHGE | 95% CI | Number of primary school children | Total emissions per year (190 school days) kgCO$_2$e |
|---|---|---|---|---|
| Pre-SFP meals | 0.72 | 0.71 to 0.74 | 1 636 833 | 223 918 754 (220 808 771 to 230 138 719) |
| SFP scenario meal | 0.79 | 0.77 to 0.81 | 1 636 833 | 245 688 633 (239 468 668 to 251 908 599) |

GHGE, greenhouse gas emission; SFP, School Food Plan.

meals which have 0.82 (0.79 to 0.84). But the difference is now reduced to 0.02 kgCO$_2$e per meal.

Similarly we analysed the data to check whether increase in sugar, salt and saturated fat could be attributable to the increased energy intake from larger meals. We estimated the mean values for these nutrients per 530 kcal meal in both scenarios. These results shows after adjusting for energy, SFP meals have similar saturated fatty acids, sugar and salt level compared with pre-SFP scenario values.

## DISCUSSION

This paper compares the nutritional quality and GHGE of primary school meals in England with and without adoption of new food-based school food standards published by the SFP. Whether the SFP meals were considered healthier than the previous situation depends on the definition of 'healthy', but when based on NMES, salt and saturated fat only, they were less healthy than previous school meals. The new meals are more greenhouse gas intensive and the total GHGE will increase by an equivalent of 220 000 economy class return journeys between London and New York.[32] There is no previous literature which has quantified these policy outcomes simultaneously in school meals.

The difference in GHGE between two scenarios is fairly small after adjusting for energy. Therefore, reformulation of primary school meals considering both nutritional quality and GHGE would allow achieving a healthy and sustainable diet. The changes to the total GHGEs of the UK food system might not be directly as estimated above due to compensatory affects later in the day. As a result of the SFP, if children get a larger meal or more red meat at school, parents may change the food they offer at home and reduce the portion sizes. We do not have data to quantify these total changes for the daily diets.

Improving both the nutritional quality and environmental impact of diets is a difficult task. Improvement in 'healthiness' of a meal does not automatically guarantee a positive environmental impact. A paper from Horgan et al[33] using adult diets from the UK National Diet and Nutrition Survey showed that it is difficult to meet all dietary recommendations and that reducing GHGEs makes it even more difficult. Tom et al[34] found that shifting current US diets to recommended food patterns based on the US Dietary Guidelines would increase energy use by 38% and GHGEs by 6%. Payne et al[17]

presented an overview of published quantitative data that indicate whether there is an association between the health and environmental impact of actual and modelled dietary patterns. They found that for salt and saturated fat, the majority of dietary patterns in the published literature found a reduction in levels of these nutrients in diets with reduced GHGEs. This may be due to the reduction in meat consumption in lower GHGE dietary patterns. Of the 12 studies that reported salt and saturated fat content of diets, eight analysed diets showed reduced levels of meat and dairy. However, the majority of dietary patterns that reported sugar intake showed increased sugar in lower GHGE diets. They also found an inconsistent relationship between reduced GHGEs and positive health outcomes.

The implementation of SFP has led to a series of new activities following its publication. In the plan 'sustainability' was mentioned under the checklist for head teachers.[22] It suggested sourcing local food where possible and using sustainable fish sources as measures to improve the sustainability of primary school meals. Previous literature has shown that changing the sources of food will not be adequate to meet the current GHGE targets set by the UK Government and there is a need to shift diets towards more sustainable diets.[8 35] Our analysis suggests that with regard to one environmental indicator—GHGEs—implementation of the SFP will not lead to a more sustainable diet and could lead to increases in emissions.

The most commonly discussed food group and the one identified as having the greatest impact in relation to the GHGE of meals is red meat.[36] The new food-based standards say 'a portion of meat or poultry must be provided at least three times each week'. Current standards do not provide any upper limit to red meat provision for children in schools despite organisations such as the World Cancer Research Fund International recommending adults to eat <500 g (cooked weight) red meat a week.[37]

The final set of standards for the SFP was published in July 2014. The Children's Food Trust had been commissioned by the SFP to develop and pilot test revised new set of food-based standards.[38] The pilot study was conducted with 35 schools and 24 caterers representing a range of different types of schools. Schools and caterers agreed that the new food-based standards were easier to follow, but thought that clear guidance and support should be provided to maintain the nutritional balance of meals. The pilot study team tested several menus,

which complied with the new standards, for their nutritional contents and concluded that meals would be the same or healthier than current school meals.[38] But these results are based on a small sample of school menus which are designed according to the SFP standards. In a real-world setting there is no guarantee that caterers would seek to create the healthiest menus that comply with the standards. They could select food items from their current menus which already comply with the new food-based standards (which is equivalent to the scenarios that we modelled in this paper). Also they can find small changes that might improve the healthiness of some meals without significant changes to the cost. One example could be replacing chips with potatoes. There are different impacts of changing menus to make them align with the SFP. It could be more sustainable and healthy. On the other hand it could contribute to more food waste, if the revised menus are less preferred and lead to larger portion sizes, which will impact on GHGEs.

There are different opinions about nutrient-based standards for school meals.[22] The pilot study conducted by the Children's Food Trust[38] found that the new food-based standards give greater flexibility for cooks to provide dishes that children like rather than having to provide a nutritionally analysed compliant menu. But some participants raised concerns that without nutrient-based standards micronutrient content would be worse and levels of fat, salt and free sugars could be higher. Similar concerns have been raised in previous reports, suggesting that although food-based standards can help to increase the intake of fruits, vegetables, oily fish, etc, they may not be sufficient to reduce the salt, sugars and fat.[39] There are several standards in the new food-based standards to reduce salt, free sugars and saturated fat.[23] Some examples are 'no confectionary at any time' and 'no salt shall be available to add to food after the cooking process is complete'. The challenge would be to maintain and monitor the salt content in cooked food, without a specific standard.

This paper shows how to quantify nutritional and GHGE outcomes of a policy option. To the best of our knowledge, this is the first study to estimate GHGE change of a national food policy. Most of the previous studies compare the GHGE of different dietary patterns and their health outcomes.[14 15 29 40]

The GHGE estimates are from a systematic review, which included papers from a wide variety of sources, countries and different production systems. This increases the range for GHGE values given for food items. This study estimated the GHGE of food items served to the plate and does not deal with the GHGE of leftover food and food waste. A study conducted in 2010 by the Waste and Resources Action Programme showed that over a school year a total of 55 000 tonnes of food is wasted by primary schools in England.[41] In this data set, we have data only for a single meal of the day, per child and it is difficult to draw conclusions about 'healthiness of the overall diets' using common tools, which require daily or weekly intakes. This paper used data from the 2009 PSFS so our models provide a comparison to this year only, this does not account for any other changes that may occur in primary school meals after this date. GHGE estimates could be influenced by changes in options available to children in schools, as a result of the new food-based dietary guidelines, along with increased uptake of free primary school meals since the introduction of universal free school meals in September 2014. Data following the introduction of the SFP were not available.

## CONCLUSIONS

The introduction of new food-based standards will increase the GHGE from primary school meals in England and improve some aspects of the nutritional quality, but it will not improve the salt, free sugars and saturated fat content levels, unless recipes are changed. A proper monitoring and evaluation framework should be in place to assess the impact of the standards on the nutritional quality of school meals. The School Food Plan Alliance and educational authorities should consider including environmental impact of food into the evaluation framework to support the Government's initiative of cutting GHGEs.

**Acknowledgements** The authors thank the Children's Food Trust (formerly the School Food Trust) for making data available for this analysis.

**Contributors** KW designed the study, created the data set for analysis, analysed data and wrote the first draft of the manuscript. PS conceptualised the study, developed the study design, supervised the analysis and revised the draft paper. MR developed the study design, revised the analysis plan and revised the drat paper. NT and MG provided inputs for the study design, commented on the analysis plan and results, revised the draft paper.

**Funding** KW and NT are supported by a grant from the British Heart Foundation (006/P&C/CORE/2013/OXFSTATS). MR and PS are supported by a programme grant from the British Heart Foundation (021/P&C/Core/2010/HPRG). MG was funded in part by Public Health England.

**Disclaimer** The views expressed are the authors' and do not necessarily reflect the views of the funding bodies.

**Competing interests** None declared.

**Provenance and peer review** Not commissioned; externally peer reviewed.

**Data sharing statement** Main data set used for this study are Primary School Food Survey data set which is available via the UK Data Archive. Please contact the corresponding author (kremlin.wickramasinghe@dph.ox.ac.uk) to access the greenhouse gas emissions data or for any further inquiries.

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
