## [Reviewer comments · BMJ Open]

ARTICLE DETAILS

TITLE (PROVISIONAL)	Environmental and nutrition impact of achieving new School Food Plan recommendations in the primary school meals sector in England
AUTHORS	Wickramasinghe, Kremlin; Rayner, Mike; Goldacre, Michael; Townsend, Nick; Scarborough, Peter

VERSION 1 - REVIEW

REVIEWER	Dr Stephen Whybrow Research Fellow Rowett Institute of Nutrition and Health University of Aberdeen Aberdeen Scotland
REVIEW RETURNED	05-Sep-2016

GENERAL COMMENTS	The manuscript of Wickramasinghe et al. describes the results of a modelling exercise looking at the likely nutritional and environmental consequences of adopting the School Food Plan. It is clear and well written, and the methods are appropriate given the available data. The manuscript adds evidence to the important message that healthy meals or diets are not necessarily more sustainable than less healthy ones. My main comment is that the analysis makes the assumption that SFP meals would be adopted as is, which would increase GHGe. The difference in GHGe between the two scenarios is fairly small (especially after accounting for the differences in meals size – line 277) and with some reformulation healthier meals could be achieved with similar, or even slightly lower GHGe. I think some discussion on this could be included without detracting from the healthy meals are not necessarily more sustainable message. Comments Line 50. “The universal adoption of the new food-based standards 50 would result in ...” This could be qualified by including “without reformulation” or similar. Would it be possible to give some information on the contribution of the various foods to the protein standard? Although red meat, white meat, fish, eggs (and alternatives?) are considered equal in the SFP, they obviously have very different associated GHGe values. This may allow the reader to see whether small menu reformulations would lower GHGe of the SFP to below the current level. For example if red meat is the main contributor to the protein standard, changing some meals to white meat or fish would lower GHGe. Line 88. “This study estimated the GHGE of food items served to the
--

	plate and does not deal with the GHGE of leftover food and food waste.” Further comment could be made in the discussion on the effects of changing menus. If more food is wasted because it is less preferred, or because of larger portion sizes, then this will impact on GHGs. Line 264. “The new meals are more greenhouse gas intensive ... New York” I think this statement needs some qualification. The SFP meals are bigger (in terms of energy) and there may be some compensatory effects later in the day that lower total daily GHGe. Red meat may be served by parents less often in the evening if it becomes more available at lunchtime. Line 309. “Current standards do not provide any upper limit to red meat provision in schools.” The World Cancer Research Fund has set recommendations for red and processed meat, although how this applies to children is not immediately clear to me. Line 320. While your point is probably true, some small changes may improve the healthiness of some meals without changing cost (much). Replacing chips with potatoes for example. Line 325. “The pilot study...” which pilot study? Minor comments. Line 47 and throughout, Kg should be kg. Line 49 and 256 “... by 22,000,000KgCO2e.” Over what time? Per day, per week, per year? Line 108. GHEs should be GHGe? Line 144. “was used in this study” is not needed. Table 1. Potatoes: 120-70. 170?
--	---

REVIEWER	Dr Stephanie Chambers University of Glasgow, UK
REVIEW RETURNED	12-Sep-2016

GENERAL COMMENTS	I enjoyed reading this paper. It presents a thought-provoking modelled comparison of the nutritional and sustainability differences between nutrient based and food based standards in English primary schools. I would recommend that this work is published as it will be of interest to readers in the UK and beyond who first of all believe that school meals have the potential to improve children's health and second, those who believe that school meals must deliver sustainability as well health benefits. The manuscript would benefit from some improvements and these are detailed below. There is a discrepancy between the aim that is in the abstract (about the method used) and the aim at the end of the introduction. These need to be reconciled. Given that there is a publication using this method published in 2016 by Wickramasinghe et al, it appears that this work is more about comparing nutrient intake and greenhouse gas emissions before and after the introduction of mandatory food based standards for school meals. Abstract - page 2, line 32 - wording should be revised to be clearer. Page 2, line 42 - not mentioned anywhere in the abstract that pre-
---

SFP means nutrient based standards. Needs to be explained.

Strengths and limitations section - page 4 - lines 84-87 - This needs to be rewritten, particularly lines 85 and 86.

Introduction - on rereading the paper I was surprised that the GHGE section (p5, lines 102-109) comes first as the results are definitely more focused around the nutritional impact rather than the carbon impact, which is only a small part of the results. The authors should consider whether they can address this in some way. A suggestion is to begin by discussing a shift in healthy eating recommendations to consider environmental impact as an introduction and then go on to bring in their info they have there already.

P5, line 114 - I was surprised that increased uptake of school meals with the introduction of universal FSM for reception, Year 1 and 2 was described as 'expected' when we know that uptake has increased. I think the increase in uptake should be reported, and in the discussion, detail the fact that this is not taken into account with the data available as a further limitation of the study.

P5, line 118 - more detail is needed on the SFP - particularly the date it was published and the date the standards came into place. This is a more general point but it could be clearer in the paper that the food based standards are mandatory. From what I understand, much of the other recommendations in the report were supported by the Secretary of State for Education, but that that food based standards were launched by the Department of Education to coincide with the publication of the School Food Plan and are mandatory. It is this that makes the standards policy rather than guidelines or recommendations and I think this distinction would be helpful.

P5, line 123 - the point from the Children's Food Trust report that nutrient standards had improved school meals is not 'in contrast' as reported. It is perfectly possible that nutrient based standards improved the nutritional quality of school meals from what they were previously, without undermining the recommendation that food based standards should now be introduced. Also, it needs to be clearer that the 'impact' described is improving nutritional value.

Methods - p6, line 144 - delete 'was used in this study'.

P7, line 151 - was information on the food items chosen only available for a subsample of children? This isn't clear. More generally, referring the reader to three different publications to get a clear idea of much of the methodology used meant that at times the method is confusing and needs very careful reading and cross checking. Word limits are always tricky to manage, but if it is possible to include more detail on how data was collected from the children, this would be beneficial. Similarly, this could be said in reference to the systematic review.

P10, line 220 - It needs to be clear earlier in the paper (in the introduction) that food based standards are being brought in instead of nutrient based standards that exist.

P10, line 232 - I was really surprised to see that only half of meals met the recommendation for not having any confectionary or chocolate. I'm more familiar with the Scottish school food environment where no confectionary can be sold, and my own

	research would suggest that compliance is very very high. Why do schools in England allow it? [more out of interest than for the paper] Discussion Whilst reading it occurred to me that given that under a food based system the options available would be quite different and therefore the eventual choices made by the children could also be different. It cannot necessarily be assumed that like for like foods would always be eaten when the range of choices is different. This should be raised as a limitation in the data. Another important limitation is that the data are from 2009 and it questionable how applicable this is to now, particularly with the increased uptake from UFSM. In the discussion the issue of increased energy intake is raised from the food based standards and a per 100g amount is calculated for GHGE. I think this should be in the results section rather than the discussion. In addition, is the increase in sugar, sat fat and salt attributable to the increased energy intake also? Given that the increases are relatively small (although I acknowledge that even small increases can contribute to excess weight etc in the longer term). Even adjusting for energy intake, SFP meals still led to increased GHGE. This is not considered further but it seems like the authors were going to say more (p12, line 269 'could be due to several reasons'), but do not address this after adjusting the energy intake. Is it more red meat, increased F&V? P 15 - I was glad to see the authors mention food waste and this is a huge problem in Scotland. Is there English data that estimates how much food is wasted? It would be good to provide an indication of this so that readers have an idea of the degree to which the data may misestimate children's intake. Tables 1 and 2 - '4-10 school years' - This is confusing as it suggests school Year 4, 5 etc - when it should be aged 4-10 years.
--	--

VERSION 1 – AUTHOR RESPONSE

Reviewer: 1

Please leave your comments for the authors below

The manuscript of Wickramasinghe et al. describes the results of a modelling exercise looking at the likely nutritional and environmental consequences of adopting the School Food Plan. It is clear and well written, and the methods are appropriate given the available data. The manuscript adds evidence to the important message that healthy meals or diets are not necessarily more sustainable than less healthy ones.

My main comment is that the analysis makes the assumption that SFP meals would be adopted as is, which would increase GHGe. The difference in GHGe between the two scenarios is fairly small (especially after accounting for the differences in meals size – line 277) and with some reformulation healthier meals could be achieved with similar, or even slightly lower GHGe. I think some discussion on this could be included without detracting from the healthy meals are not necessarily more sustainable message.

Thank you for your comments. We have addressed this comment in the discussion and also

responded to other comments as described below.

We have added a paragraph which reads as- (Page 13 Line 295). . Please note that my page numbers and line numbers in the response letter refers to the corresponding numbers in the version with track changes.

“The difference in GHGE between two scenarios is fairly small after adjusting for energy. Therefore reformulation of primary school meals considering both nutritional quality and GHGE would allow to achieve a healthy and sustainable diet. The changes to the total GHGEs of the UK food system might not be directly as estimated above due to compensatory affects later in the day. As a result of the SFT, if children get a larger meal or more red meat at school, parents may change the food they offer at home and reduce the portion sizes. We do not have data to quantify these total changes for the daily diets.”

Comments

Line 50. “The universal adoption of the new food-based standards 50 would result in ...” This could be qualified by including “without reformulation” or similar. –

(Page 3 Line 55). Sentence was revised as “The universal adoption of the new food-based standards, without reformulation would result in an increase in the GHGEs of school meals and improve some aspects of the nutritional quality, but it would not improve the average salt, sugar and saturated fat content levels, unless recipes are changed.”

Would it be possible to give some information on the contribution of the various foods to the protein standard? Although red meat, white meat, fish, eggs (and alternatives?) are considered equal in the SFP, they obviously have very different associated GHGe values. This may allow the reader to see whether small menu reformulations would lower GHGe of the SFP to below the current level. For example if red meat is the main contributor to the protein standard, changing some meals to white meat or fish would lower GHGe.

We agree this could have provided more insights in to our findings. But due to the way our dataset is organised, we are unable to present information on the contribution of the various foods to the individual nutrients including protein.

In this dataset, we had individual food items consumed by individual children. We could assign each food item to a food group such as red meat, white meat, fish and eggs.

To analyse protein (and other nutrient) levels in each meal, it is important to define “individual meal” from the database (rather than individual food items). The database lists individual food items consumed by a student separately. The dataset list detail of individual selection of food items by students in their plates for lunch with their nutritional values for 14 nutrients. It also contains the school id and student id with each food item. We sorted data by the student ID variable, to understand what comprised an individual school lunch consumed by study participants and to define an individual school meal to understand their overall nutritional value. (E.g. when we sorted by student id 1 of the school 100260, the individual student’s lunch on that day comprised items shown in the table below.

Table: For Reviewer’s Comment: List of food items selected by an individual student

Student ID	School ID	age	Day of the week	food	Food code	Seconds?
1	100260	5	Monday	Flapjack	84	No
1	100260	5	Monday	Custard	109	No
1	100260	5	Monday	Cucumber	266	No
1	100260	5	Monday	Sweetcorn	309	No
1	100260	5	Monday	Cup of tap water	539	No
1	100260	5	Monday	Potato	799	No

Then we collapsed the dataset, using STATA, by individual student ID, to get the sum of nutritional values for each nutrients. At this stage we lose the individual food code value and the food group value assigned to each food item. We had more than 33,000 observations (food items) and it is not possible at this stage to change the dataset or to trace the contributions from unique food items to each nutrient in individual meals. Therefore, we are unable to analyse contribution of the various foods to the individual nutrients.

Line 88. "This study estimated the GHGE of food items served to the plate and does not deal with the GHGE of leftover food and food waste." Further comment could be made in the discussion on the effects of changing menus. If more food is wasted because it is less preferred, or because of larger portion sizes, then this will impact on GHGs.

The discussion section is now updated to say

(Page 15 Line 359) "There are different impacts of changing menus to make them align with the SFP. It could be more sustainable and healthy. On the other hand it could contribute to more food waste, if the revised menus are less preferred and lead to larger proportion sizes, which will impact on GHGs."

Line 264. "The new meals are more greenhouse gas intensive ... New York" I think this statement needs some qualification. The SFP meals are bigger (in terms of energy) and there may be some compensatory effects later in the day that lower total daily GHGe. Red meat may be served by parents less often in the evening if it becomes more available at lunchtime.

A sentence was added to address this.

(Page 13 Line 297) "The changes to the total GHGs of the UK food system might not be directly as estimated above due to compensatory effects later in the day. As a result of the SFT, if children get a larger meal or more red meat at school, parents may change the food they offer at home and reduce the portion sizes. We do not have data to quantify these total changes for the daily diets"

Line 309. "Current standards do not provide any upper limit to red meat provision in schools." The World Cancer Research Fund has set recommendations for red and processed meat, although how this applies to children is not immediately clear to me.

We have checked again, and according to our understanding there are no upper limit for red meat set for children. We have updated the text to reflect the above point and this section now reads as:

(Page15 Line 339)"The most commonly discussed food group and the one identified as having the greatest impact in relation to the GHGE of meals is red meat [33]. The new food based standards say "a portion of meat or poultry must be provided at least three times each week". Current standards do not provide any upper limit to red meat provision for children in schools despite organisations such as the World Cancer Research Fund International recommending adults to eat less than 500g (cooked weight) red meat a week [39]."

Line 320. While your point is probably true, some small changes may improve the healthiness of some meals without changing cost (much). Replacing chips with potatoes for example.

The text is now updated to include this point. It now reads as

(Page 15 Line 355) "They could select food items from their current menus which already comply with the new food-based standards (which is equivalent to the scenarios that we modelled in this paper).

Also they can find small changes that might improve the healthiness of some meals without significant changes to the cost. One example could be replacing chips with potatoes.”

Line 325. “The pilot study...” which pilot study?

We have added the reference and now this line reads as

(Page 16 Line 365)“There are different opinions about nutrient based standards for school meals. [19]. The pilot study conducted by the Children’s Food Trust found that...”

Minor comments.

Line 47 and throughout, Kg should be kg. This is now changed as kg

Line 49 and 256 “... by 22,000,000KgCO₂e.” Over what time? Per day, per week, per year?

This sentence now reads as “Adopting the SFP would increase the total emissions associated with primary school meals by 22,000,000kgCO₂e per year.”

Line 108. GHEs should be GHGe?

This is changed and now reads as (Page 5 Line 112) “There is a growing literature looking at the GHGE of diets and their nutritional quality or health impacts [7 10-13].”

Line 144. “was used in this study” is not needed.

This is now deleted. The sentence now reads as

(Page 7 Line 153) “This study used the “Primary School Food Survey 2009 (PSFS)” data set, a nationally representative survey in 139 primary schools in England [21]”

Table 1. Potatoes: 120-70. 170?

This is now changed as “120-170”

Reviewer: 2

I enjoyed reading this paper. It presents a thought-provoking modelled comparison of the nutritional and sustainability differences between nutrient based and food based standards in English primary schools. I would recommend that this work is published as it will be of interest to readers in the UK and beyond who first of all believe that school meals have the potential to improve children's health and second, those who believe that school meals must deliver sustainability as well health benefits.

The manuscript would benefit from some improvements and these are detailed below.

There is a discrepancy between the aim that is in the abstract (about the method used) and the aim at the end of the introduction. These need to be reconciled. Given that there is a publication using this method published in 2016 by Wickramasinghe et al, it appears that this work is more about comparing nutrient intake and greenhouse gas emissions before and after the introduction of mandatory food based standards for school meals.

Thank you for this comment. We had changed the aim in the abstract now to read as “The aim of this modelling study was to estimate the expected changes in the nutritional quality and GHGEs of primary school meals due to the adoption of new mandatory food-based standards for school meals.”

Abstract - page 2, line 32 - wording should be revised to be clearer.

Thank you for this comment. The sentence now reads as:

(Page 2 Line 31) “Setting: Nationally representative random sample of 136 primary schools in England, was selected for the Primary School Food Survey (PSFS) with 50% response rate.”

Page 2, line 42 - not mentioned anywhere in the abstract that pre-SFP means nutrient based standards. Needs to be explained.

(Page 2 Line 34) We have now included a sentence to the abstract to explain this point. The new sentence is “ Prior to introduction of the SFP (pre- SFP scenario), the primary school meals had mandatory nutrient based guidelines to be achieved.”

Strengths and limitations section - page 4 - lines 84-87 - This needs to be rewritten, particularly lines 85 and 86.

The section is now updated and the above mentioned point has been edited as below:

(Page 4 Line 85))

“The GHGEs estimates are from a systematic review conducted by authors (for a previous study). Since GHGE per food item comes from multiple sources, it increases the range for GHGE values given for food items compared to studies which take them from a single source. These reported GHGE values are more acceptable given that it is very difficult to pick a single study to best represent the GHGE values for a range of different sources of food.”

Introduction - on rereading the paper I was surprised that the GHGE section (p5, lines 102-109) comes first as the results are definitely more focused around the nutritional impact rather than the carbon impact, which is only a small part of the results. The authors should consider whether they can address this in some way. A suggestion is to begin by discussing a shift in healthy eating recommendations to consider environmental impact as an introduction and then go on to bring in they info they have there already.

We added a new first paragraph and changed the introduction (Page 5 Line 101). For example the first paragraph now reads “Schools are an important partner in population level nutrition promotion [1]. Over the last few decades there has been many policies to promote diet and physical activity in schools [2]. Recent policies have addressed the issue of sustainability of school food in addition to their nutritional quality [3].”

P5, line 114 - I was surprised that increased uptake of school meals with the introduction of universal FSM for reception, Year 1 and 2 was described as 'expected' when we know that uptake has increased. I think the increase in uptake should be reported, and in the discussion, detail the fact that this is not taken into account with the data available as a further limitation of the study.

Thank you for this comment. We deleted the sentence “It is expected that the uptake of primary school meals will increase significantly with the new policy” and added a new sentence to confirm that the uptake of free school meals has gone up since the introduction of universal FSM for the above age group. It reads as (Page 5 Line 118) “Official figures show that within three months of introducing this policy 1.3 million more children started eating free school meals, resulting in an uptake close to 85% of all infants”.

Also a sentence was added to the discussion “These results do not reflect the increase in uptake of free primary school meals since the introduction of universal free school meals in September 2014.”

P5, line 118 - more detail is needed on the SFP - particularly the date it was published and the date the standards came into place. This is a more general point but it could be clearer in the paper that the food based standards are mandatory. From what I understand, much of the other recommendations in the report were supported by the Secretary of State for Education, but that that food based standards were launched by the Department of Education to coincide with the publication of the School Food Plan and are mandatory. It is this that makes the standards policy rather than guidelines or recommendations and I think this distinction would be helpful.

We have updated the section to clarify this comment. This section now reads as (Page 6 Line 123) “One of the most recent changes to school meals in England was introduced by the School Food Plan

(SFP) [22]. It provided a series of action points for head teachers to improve the quality of school meals. Based on the suggestions of an expert panel appointed to review school food standards in the UK, the SFP concluded that nutrient-based standards were too difficult to interpret and school meal guidelines should be based only on food-based standards [23]. A report of the Children's Food Trust had found that the introduction of nutrient based standards had improved school meals in England [24]. There has long been a division of opinion amongst nutrition experts regarding whether food based standards are more beneficial than nutrient based standards [25]. A new set of food-based standards (instead of nutrient based standards) was published by the Department for Education in June 2014. Since September 2014, schools are now legally required to provide meals that comply with these standards [26]. The impact of these food based standards is also to improve the nutritional values of school meals.”

P5, line 123 - the point from the Children's Food Trust report that nutrient standards had improved school meals is not 'in contrast' as reported. It is perfectly possible that nutrient based standards improved the nutritional quality of school meals from what they were previously, without undermining the recommendation that food based standards should now be introduced. Also, it needs to be clearer that the 'impact' described is improving nutritional value.

(Page 6 Line 128) We have deleted the word “in contrast and added a sentence to describe that even with the food based guidelines the expected impact is to improve nutritional values of primary school meals. Please see the response in the above comment, this comment is also addressed within the same paragraph.

Methods - p6, line 144 - delete 'was used in this study'.

This is deleted. This sentence now reads as (Page 7 Line 153) “This study used the “Primary School Food Survey 2009 (PSFS)” data set, a nationally representative survey in 139 primary schools in England [21]”

P7, line 151 - was information on the food items chosen only available for a subsample of children? This isn't clear. More generally, referring the reader to three different publications to get a clear idea of much of the methodology used meant that at times the method is confusing and needs very careful reading and cross checking. Word limits are always tricky to manage, but if it is possible to include more detail on how data was collected from the children, this would be beneficial. Similarly, this could be said in reference to the systematic review.

Information on food items chosen are available for all the students who are included in the PSPFS survey. In this sentence the phrase “a sample of students”, which comes from PSFS report, refers to the sample of students selected from each school, rather than a subsample of the study participants. To avoid this confusion, we have restructured the paragraph. Now it reads as “The unit in the PSFS dataset is a food item (i.e. individual foods that combine to form a meal, e.g. ‘spaghetti’ and ‘bolognese sauce’). Nutritional information for all the foods available to the student on that day and food items chosen by all the participants of the PSFS survey was recorded in the PSFS dataset. Nutritional content of the foods was estimated with reference to the Food Standards Agency Nutrient Databank [24]. In the PSFS dataset there were 1,556 unique food items consumed by students.”

The total word limit already comes more than 3,500 words, we therefore did not provide any more additional details on methods. We hope we have provided essential key information along with reference for further details that readers are able to access. We are happy to add more if more space could be provided by the journal editors for the article.

P10, line 220 - It needs to be clear earlier in the paper (in the introduction) that food based standards are being brought in instead of nutrient based standards that exist.

This is now added to the introduction. The new sentence reads (Page 6 Line 132) "A new set of food-based standards (instead of nutrient based standards) was published by the Department for Education in June 2014. Schools are now legally required to provide meals that comply with these standards [23]. The impact of these food based standards is also to improve the nutritional values of school meals."

P10, line 232 - I was really surprised to see that only half of meals met the recommendation for not having any confectionary or chocolate. I'm more familiar with the Scottish school food environment where no confectionary can be sold, and my own research would suggest that compliance is very very high. Why do schools in England allow it? [more out of interest than for the paper]

Thank you for this comment. According to the Primary School Food Survey report this is now improving. Also the School Food Plan has a specific point on this and highlights it as a check list item for head teachers. We hope this recommendation will be met by most schools with the new guidelines and we will have to wait for new data to find out the current status.

Discussion

Whilst reading it occurred to me that given that under a food based system the options available would be quite different and therefore the eventual choices made by the children could also be different. It cannot necessarily be assumed that like for like foods would always be eaten when the range of choices is different. This should be raised as a limitation in the data.

We have added this to the limitations section. Now it reads as (Page 17 Line 394)

"This paper used data from the 2009 PSFS so our models provide a comparison to this year only, this does not account for any other changes that may occur in primary school meals after this date. GHGE estimates could be influenced by changes in options available to children in schools, as a result of the new food-based dietary guidelines, along with increased uptake of free primary school meals since the introduction of universal free school meals in September 2014. Data following the introduction of the SFP were not available."

Another important limitation is that the data are from 2009 and it questionable how applicable this is to now, particularly with the increased uptake from UFSM.

Please see the response for the comment above. This point is now mentioned in the new limitation paragraph.

In the discussion the issue of increased energy intake is raised from the food based standards and a per 100g amount is calculated for GHGE. I think this should be in the results section rather than the discussion. In addition, is the increase in sugar, sat fat and salt attributable to the increased energy intake also? Given that the increases are relatively small (although I acknowledge that even small increases can contribute to excess weight etc in the longer term).

We have removed the above mentioned paragraph from the discussion section now and added this point to the results section. The new paragraph of the results now read as (Page 11 Line 268 onwards):

"The difference in the mean GHGEs in the SFP and pre-SFP scenarios could be associated with the amount of food items included in the meal, which makes it more likely to achieve food based SFP standards and therefore with the total energy of the meal. To test this, we estimated the GHGE per set amount of energy in a primary school meal. The mid-point of the energy reference range for primary school meals is 530Kcal. We estimated the GHGE for a 530Kcal meal in SFP scenario and in

a meal in the Pre-SFP scenario.

When GHGE per 530 Kcal is estimated in SFP- and Pre-SFP meals, the SFP meals still have slightly more GHGEs 0.84 (0.81 – 0.86) compared to the pre-SFP meals which have 0.82 (0.79 – 0.84). But the difference is now reduced to 0.02 KGCO₂e per meal.”

We have analysed the data to check whether increase in sugar salt and saturated fat could be attributable to the increased energy intake from larger meals. We estimated the mean values for these nutrients per 530Kcal meal in both scenarios. These results shows after adjusting for energy, SFP meals have similar saturated fatty acids, sugar and salt level compared to pre-SFP scenario values.

Even adjusting for energy intake, SFP meals still led to increased GHGE. This is not considered further but it seems like the authors were going to say more (p12, line 269 'could be due to several reasons'), but do not address this after adjusting the energy intake. Is it more red meat, increased F&V?

We have edited this discussion section and moved the content to the results section as described in the response to the comment above.

We have explained in a response to first reviewer that according to the format of the database we are unable to estimate the contribution of each food item to the GHGE value or nutrient value of individual meals. Please see our response to the first reviewer’s comment regarding “contribution of the various foods to the protein standard?”.

It is difficult to provide the exact reason for this increase GHGE based on our data and we did not want to provide any reasons that cannot be supported by our findings. We have provided the energy adjusted values in the results section and it provides some explanation.

P 15 - I was glad to see the authors mention food waste and this is a huge problem in Scotland. Is there English data that estimates how much food is wasted? It would be good to provide an indication of this so that readers have an idea of the degree to which the data may misestimate children's intake. It is difficult to estimate the exact percentage of food waste from primary school meals and the GHGEs associated with this waste. We have quoted a figure from a study conducted in 2010. This is now mentioned and referenced in the limitations section. It now reads as (Page 16 Line 385) “This study estimated the GHGE of food items served to the plate and does not deal with the GHGE of leftover food and food waste. A study conducted in 2010 by the Waste and Resources Action Programme showed that over a school year a total of 55,000 tonnes of food is wasted by primary schools in England”

Tables 1 and 2 - '4-10 school years' - This is confusing as it suggests school Year 4, 5 etc - when it should be aged 4-10 years.

This is now updated as “aged 4- 10 years”

VERSION 2 – REVIEW

REVIEWER	Stephen Whybrow Rowett Institute University of Aberdeen Foresterhill Aberdeen Scotland
REVIEW RETURNED	06-Dec-2016

GENERAL COMMENTS	My questions and comments have been addressed.
--

REVIEWER	Stephanie Chambers University of Glasgow, Scotland
REVIEW RETURNED	20-Dec-2016

GENERAL COMMENTS	I have no further issues on this manuscript and I am happy to recommend its acceptance for publication.
---